# Metaplasia of respiratory and digestive tissues in the eastern oyster *Crassostrea virginica* associated with the Deepwater Horizon oil spill

**Deanne S. Roopnarine**[1☯]*, **Peter D. Roopnarine**[2☯], **Laurie C. Anderson**[3☯], **Ji Hae Hwang**[4☯], **Swati Patel**[1☯]

**1** Department of Biological Sciences, Nova Southeastern University, Ft. Lauderdale, Florida, United States of America, **2** Department of Invertebrate Zoology and Geology, California Academy of Sciences, San Francisco, California, United States of America, **3** Department of Geology and Geological Engineering, Museum of Geology, South Dakota School of Mines and Technology, Rapid City, South Dakota, United States of America, **4** Kent State University College of Podiatric Medicine, Independence, Ohio, United States of America

☯ These authors contributed equally to this work.
* roopnari@nova.edu

**Data Availability Statement:** The data consist of oyster shells and histological tissue slides. The former will be available in the geological collections

## Abstract

Metaplasia is a well documented and deleterious effect of crude oil components on oysters. This reversible transformation of one cell type to another is a common response to petroleum-product exposure in molluscs. It has been shown experimentally in previous work that eastern oysters (*Crassostrea virginica*) exposed to petroleum products will exhibit metaplasia of digestive tissues. Here we document for the first time that wild adult oysters inhabiting coastal waters in the northern Gulf of Mexico during and in the aftermath of the Deepwater Horizon oil spill (2010) exhibited metaplasia in both ctenidial (respiratory and suspension feeding) and digestive tract tissues at significantly higher frequencies than geographic controls of *C. virginica* from Chesapeake Bay. Metaplasia included the loss of epithelial cilia, transformations of columnar epithelia, hyperplasia and reduction of ctenidial branches, and vacuolization of digestive tissues. Evidence for a reduction of metaplasia following the oil spill (2010-2013) is suggestive but equivocal.

## Introduction

The Deepwater Horizon (DWH) petroleum drilling rig suffered an explosion on April 20th, 2010, resulting in loss of the rig and rupturing of the Macondo well-head on the seafloor at a depth of 1522 m. During the next four months it is estimated that approximately 4 million barrels of oil were released into the Gulf of Mexico (GoM) [1], making the DWH spill the world's largest accidental spill in history. In addition, about 6.97 million liters of dispersant, a mix of surfactants and hydrocarbon solvents, were applied both at the wellhead and on surface slicks during the course of remediation efforts [2]. Sensitivity of coastal environments to spill

of the South Dakota School of Mines and
Technology. Tissue slides are deposited in the
Invertebrate Zoology collections of the California
Academy of Sciences. Catalog numbers for
specimens at both institutions are included in the
manuscript, and repeated here: CASIZ 234232-
234238 and SDSM 157561-157567.

**Funding:** D. R. Grant No.418335317 The Nova
Southeastern University President's Faculty
Research and Development https://www.nova.edu/
academic-affairs/faculty-research-grant/index.html
L. C. A. No grant number. Louisiana Sea Grant
https://www.laseagrant.org/.

**Competing interests:** The authors have declared
that no competing interests exist.

contamination, especially salt marshes and oyster reefs, were of grave concern because of their importance to commercial fisheries, their critical role as a line of defense against coastal erosion, and because they are more difficult to 'clean' than barrier-island beaches [3, 4].

The immediate impacts of the spill were evident, including oil slicks, fouled beaches in the GoM, and fouled, often dead wildlife; longer-term impacts are less well understood, are still being documented, and/or remain equivocal or variable. For example, studies of the incorporation of polycyclic aromatic hydrocarbons (PAHs) into a broad array of marine organisms, including oysters, crustaceans, and fish, reported elevated levels in edible tissues through 2010, with declining levels thereafter through August 2011 [5, 6]. Small particulate and mesozooplankton assemblages collected from Mobile Bay, Alabama from August to October 2010, exhibited isotopically lighter $\delta^{13}C$ compositions, indicative of oil incorporation into tissues, and therefore into the base of the GoM coastal food web [7], although the disturbance to community structure was transient and brief [8]. During 2010, the food web transitioned from metazoan dominated benthic communities to fungal dominated assemblages [9], yet during the same interval, there was little change to $\delta^{13}C$ and $^{14}C$ content of soft-tissues in the marsh mussel *Geukensia demissa* and balanoid barnacles in the Barataria Bay estuarine region of the Mississippi River delta [10]. Furthermore, Carmichael et al. [11] did not find evidence of oil-derived carbon or nitrogen incorporation into shell organic matrix or soft tissues of the Atlantic oyster *Crassostrea virginica* based on $\delta^{13}C$ and $\delta^{15}N$ values.

The extensive use of dispersants in this spill, and their potential persistence in coastal areas (see [2]), also may have impacted regional ecosystems. The acute toxicity of anionic surfactants such as dioctyl sodium sulfosuccinate (DOSS) and the commercial oil-disperant products incorporating them is well documented for aquatic organisms, particularly fishes ([12] and references there in) and dispersant may have accelerated uptake of petroleum compounds by organisms exposed to oil in this spill [7]. In addition, experimental work by Bodinier et al. [12] links DOSS exposure to acute toxicity (including death) in the Gulf killifish (*Fundulus grandis*), with increased toxicity at salinities deviating the most from isosmotic values.

Ecological impacts beyond immediate and near-term mortality caused by fouling and toxic effects of oil, its components, dispersant, and dispersate-associated oil, are poorly constrained in general. Remediation efforts included large-scale diversion of Mississippi River water into Louisiana coastal wetlands during the summer of 2010 at a combined maximum flow of 780 $m^3$/sec (Louisiana Office of Coastal Protection and Restoration). The diversions, particularly outflow of the Caernarvon Freshwater Diversion into Breton Sound and Davis Pond and eventually into northern Barataria Bay, altered salinity regimes and nutrient levels [13, 14], and likely the hydrodynamics of coastal marshes and nearshore waters. This sustained freshwater influx may also have had adverse effects on oysters, as the release occurred during spawning season and declines in oyster abundance, spat settlement, and filtration rates are associated with reduced salinities caused by flooding [15]. Indeed, oyster harvests on public leases were delayed east of the Mississippi River in 2010 due to a depressed spat set and oyster mortalities [16]. In addition, dramatic density decreases of oysters in areas affected by freshwater diversion efforts have been documented [17, 18], although in experimental work, Schrandt et al. [19] found that lower salinity conditions (5-10 ppt) led to increased survival of juvenile *C. virginica* exposed to oil and dispersed oil. In addition, Dietl and Durham [20] did not detect any significant difference of adult shell size between a baseline of pre-spill historical specimens and those collected in the years 2011 to 2013.

An expectation of delayed effects, and long-term consequences, of the DWH blowout is supported by studies of the Exxon Valdez spill, where both ecological and physiological impacts on a variety of marine organisms continued decades after the event [21]. An important control on any assessment of the DWH impact, however, is the establishment of the state

of the GoM prior to the spill. The GoM is home to more than 4,000 offshore production plat-forms, with the first offshore production dating to 1937, and thousands more platforms being added in the ensuing decades [3, 22]. There have been at least five spills exceeding one million barrels prior to the DWH in the GoM, and ongoing lesser incidents and leakages (NOAA Office of Response and Restoration). There are also numerous natural, subtidal hydrocarbon seeps along the GoM continental shelf and slope ($\sim$ 140,000 metric tons oil/gas per year released into the northern GoM [23]). Organisms in coastal waters of the GoM have therefore likely been exposed to naturally-seeped hydrocarbons for millenia, at least since sea level rise caused by the last glacial maximum, and to hydrocarbons introduced by drilling and associated activities over the last eight decades. Additional factors complicating evaluation of spill effects include: (a) increased loads of agricultural and industrial chemicals carried into the Gulf by the Mississippi River since the 20th century [24–26], and (b) fisheries methods commonly in use in the northern GOM that can profoundly alter biotic and abiotic environmental condi-tions [27–29].

Historical baselines of conditions within the GoM would, therefore, be useful for the proper attribution of altered conditions to DWH. Reconstructing historical baselines, however, can be limited by the availability of suitable materials, and this is particularly acute in cases where analyses depend on fresh biological materials. An alternative to reconstructing the pre-DWH state of the GoM is to substitute space for time with comparative studies of the GoM to other geographical areas among which environmental factors vary.

The purpose of the study reported here was to examine possible impact of the DWH spill on soft-tissue morphology of the commercially significant American oyster *Crassostrea virgi-nica*, and to monitor its continuing or diminishing effects for a period of three years after DWH, comparing specimens collected within the GoM during that time period as well as from Chesapeake Bay in Maryland, another area where *C. virginica* is subjected to anthropo-genic impacts [30, 31], but without petroleum-based impacts on the scale of the Gulf of Mexico. Metaplasia, the reversible transformation of one cell type to another has been recorded previously in molluscs exposed to petroleum contamination [32–38] and, therefore, may occur in individuals exposed to DWH oil. We hypothesized that exposure to the DWH spill led to metaplasia of ctenidial (gill) and digestive tissues, although the timescale of reversal in individuals would remain unknown because of the destructive nature of individual sampling.

Evidence of DWH impact on *C. virginica* to date have been either negative or equivocal. Soniat et al. [39] recorded insignificant PAH and parasitic infection levels in tissues six months after termination of the spill from individuals collected east of the Mississippi River in Louisi-ana, although there is no evidence that those specimens were directly exposed to spilled oil. Preliminary measures of heavy metal concentrations in *C. virginica* shells and tissues [40] from coastal areas of Louisiana and Alabama yielded marginally significant or insignificant values, due possibly to the low metal content of DWH oil [41] or the short residence time of some key metals in oyster tissues [42]. Likewise, linking metaplasia in other organisms to DWH has been equivocal. For example, Bentivegna et al. [43] noted a significantly higher occurrence of various lesions and metaplasia in menhaden (*Brevoortia*) from Barataria Bay compared to individuals from Delaware, but were unable to establish a significant relationship to PAH levels and the DWH spill.

Experimental exposures of bivalved molluscs to crude oil derivatives, however, have resulted in a variety of pathological responses, dependent on the contaminants (metals, hydro-carbons, etc.), including lesion development, tissue necrosis and metaplasia of gill, digestive, and renal tissues [32, 37]. Vignier et al. [44] showed that *C. virginica* larvae are capable of ingesting oil that has adhered to phytoplankton on which the larvae feed, with negative effects

on larval survival, and that settled spat suffered metaplastic alterations of digestive tissues [38].
In vivo monitoring has also associated metaplasia with petroleum-based and other pollutants,
including PCBs [35]. NOAA's long-term Mussel Watch biomonitoring program [45] ranks, in
order of decreasing occurrence of pathologies: metals, pesticides and PAHs [46].

The duration and frequency of metaplasia may also be indicators of the longevity of a spill's
impact. For example, the mussel *Mytilus trossulus* in Prince William Sound continued to
exhibit metaplasia of the digestive gland more than five years after the Exxon Valdez spill in
direct correspondence to PAH concentrations [47, 48]. Furthermore, whereas metaplasia is by
definition reversible, some mollusc populations might adapt evolutionarily to chronic long-
term exposure. For instance, populations of *Caryocorbula caribaea* living near active hydrocar-
bon seeps in Trinidad, are significantly more tolerant of PAH exposure than those living in
non-seep areas [49]. Even seep-associated individuals of this species, however, exhibited
pathologies when PAH concentrations were elevated experimentally above ambient seep con-
ditions, including metaplasia of stomach epithelia.

More broadly, metaplasia may be a common molluscan response to contaminant exposure.
For example, Anitha et al. [50] reported damage to the adductor muscle, mantle and gill tissues
of *Crassostrea madrasensis* after exposure to copper. However, they noted that the most delete-
rious effects were to the epithelial tissues of the organism, noting a loss of cilia in the gills as
well as vacuolization. Calabrese et al. [51] reported the same in the digestive diverticula of the
blue mussel *Mytilus edulis*. Berthou et al. [52] studied oyster tissue after the Amoco Cadiz oil
spill in 1978 and noted changes to gonad, gill and digestive tract epithelia the year of the spill
with a return to normal morphology after three to five years. The changes included atrophy
and degeneration of tissues, particularly in the digestive tract.

Reversal, acclimatization, and adaptation thus all complicate the assessment of metaplasia
and other tissue pathologies in response to major exposures to crude oil and petroleum prod-
ucts. Here we used a comparative approach to explore the occurrence and areas of metaplasia
in specimens of *C. virginica* from the Gulf of Mexico which would have been exposed to the
Deepwater Horizon spill, plus individuals that survived the spill, and individuals from Chesa-
peake Bay in Maryland which were never exposed to the spill. The major objectives of this
study were to: (1) determine the types of epithelia lining the ctenidia and intestinal tract of *C.
virginica* and establish whether there is evidence of metaplasia; (2) compare specimens from
the Gulf of Mexico and Chesapeake Bay; (3) test whether there was a significantly elevated fre-
quency of metaplasia in specimens from the GoM, and if so, (4) determine if the frequency
varied during the time period covered by the study (2010-2013). Ctenidia and digestive tissues
were selected because of their direct interaction with suspended food particles and potentially
greater exposure to assimilated hydrocarbon materials. These tissues also play critical roles in
the health of individuals and populations.

The ctenidia of *C. virginica* are typical lamellibranch gills, consisting of branched, ciliated
filaments, and are responsible for suspension feeding and gas exchange. The particular lamelli-
branch branching morphology, ciliary arrangements, morphologies and subsequent current
system vary among lamellibranch species, but in general the ciliated linings are constructed of
one to two rows of simple, columnar epithelial cells (for example, Fig 1A). As in other ostreids,
the ctenidial filaments of *C. virginica* are borne on prominent folded ridges (plicate ctenidia),
and comprise ciliated, columnar epithelia. Ctenidial columnar cells have been observed to be
underlain by cuboidal and sometimes flattened cells [53], and in the case of the oyster *Ostrea
chilensis*, arise during development from ectodermal ridges on the mantle [54].

We report below that the digestive system of *C. virginica* is typical of bivalves, consisting of
a stomach bearing a crystalline style used in the mastication of food particles, and an upper

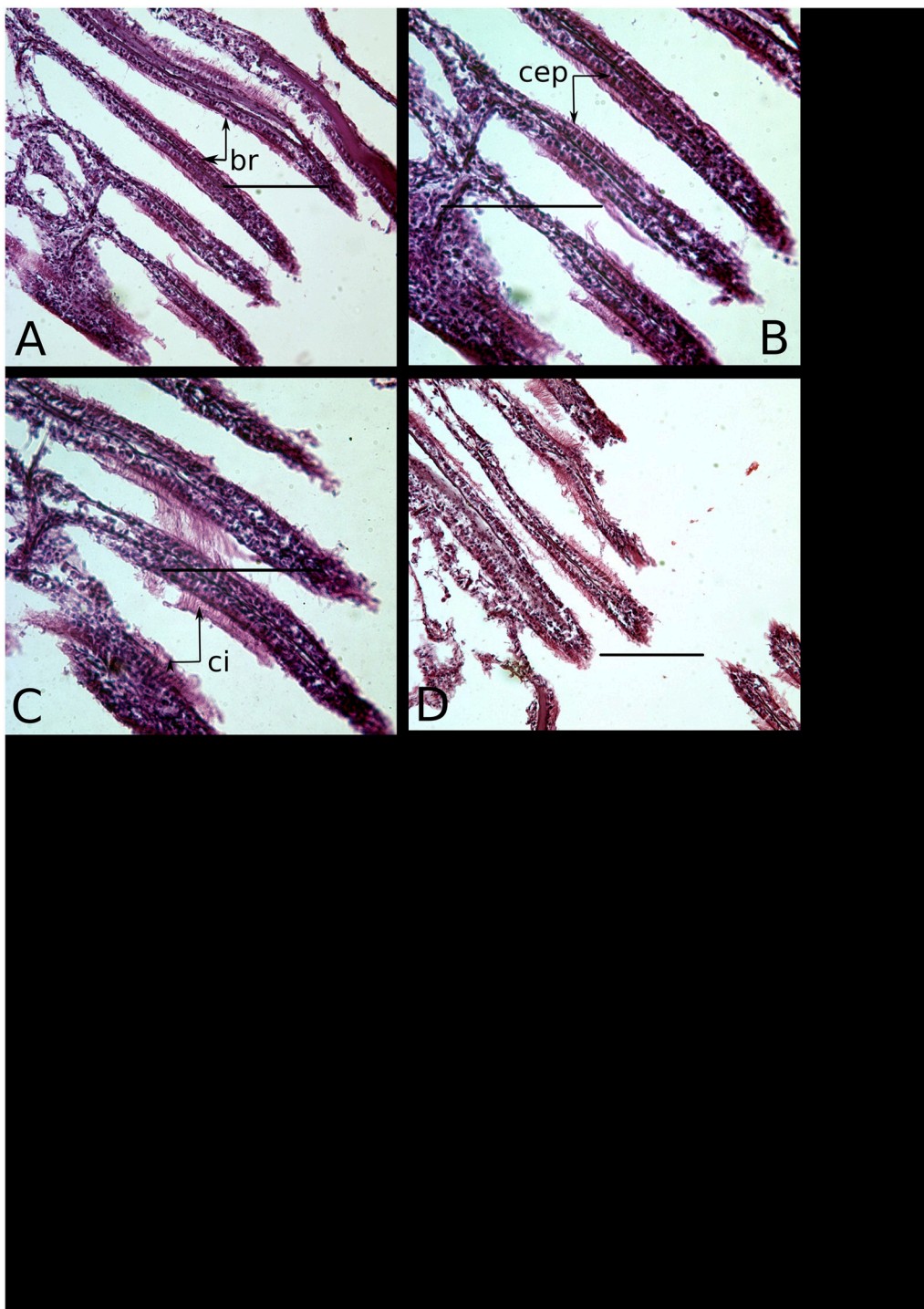

**Fig 1. Normal ctenidial branches of *C. virginica* individuals from Chesapeake Bay.** Three individuals are represented, the first in photographs A and B, and then C and D respectively. Labels: br—ctenidial branch; ci—cilia; cep—columnar epithelia. All scale bars are 100$\mu$m.

digestive tract bearing internal, ciliated branches. Those branches presumably provide increased surface area for the absorption of digested material.

## Materials and methods

### Specimen material

A total of 38 specimens from the Gulf of Mexico (GoM) and eight control specimens from Chesapeake Bay, Maryland, were selected for analysis. GoM specimens included those collected after initiation of the oil spill in 2010, and in subsequent years 2011 and 2013 (Tables 1 and 2). Oysters were collected in shallow subtidal areas within or immediately adjacent to *Spartina* marshes in a back-barrier lagoon connected to Barataria Bay at the eastern end of Grand Isle, Louisiana (LA) (29° 15.913' N, 89° 57.187' W) and near the Alabama (AL) mainland along the causeway to Dauphin Island, AL (30° 20.222' N, 88° 7.786' W) in August 2010, two months after oil made landfall. Additional oysters were collected at Grand Isle in June 2013. Control specimens from Chesapeake Bay (37° 11.953' N, 37° 11.4' W) were obtained in years 2010, 2012 and 2013. Oyster clusters were collected via visual searches until at least 10 live individuals were found. Additional oysters were obtained fresh from oyster fishermen collecting in Apalachicola Bay, FL in December 2010 and December 2011 (29° 39.772' N, 84° 59.967' W). Specimens from all localities were collected between May and August of the sampled years. Shells ranged in height between 8-12 cm, ensuring that all individuals were postjuveniles. Oysters were placed on ice immediately after collection and stored below 0°C. Specimens were subsequently frozen and stored at -18° to -20°C until retrieved for analysis. Storage times varied from 3 days to 16 months.

Freezing as a preservation technique for histological analysis can be problematic because of the formation of ice crystals at the cellular level, and the consequential damage of tissues. Specimens for this study were frozen because they were originally intended for chemical analysis of soft and hard tissues, specifically the measurement of heavy metal concentrations [40]. Thus chemical preservation was avoided and freezing employed instead. We tested for the possible introduction of histological artifacts due to freezing, and the possibility of attributing those incorrectly to metaplasia, by freezing the control specimens from Chesapeake Bay at temperatures and for durations comparable to those used for the GoM specimens.

**Table 1. Specimens and ctenidial examinations.** n is the number of specimens from each locality and year successfully examined. Ciliated—number of specimens with ciliated ctenidial branches. Epithelial condition—condition of ctenidial branch epithelia. Columnar epithelia are the normal condition, cuboidal and squamous (stratified) are metaplasial conditions. A single metaplasia specimen may possess multiple epithelial types.

| Locality | Year | n | Metaplasia | Ciliated | Epithelial condition | | |
|----------|------|---|------------|----------|----------|----------|----------|
| | | | | | columnar | cuboidal | squamous |
| Apalachicola | 2010 | 7 | 7 | 0 | 0 | 0 | 7 |
| | 2011 | 4 | 4 | 0 | 0 | 0 | 4 |
| | 2013 | 9 | 4 | 4 | 5 | 2 | 2 |
| Chesapeake | 2010 | 1 | 0 | 1 | 1 | 0 | 0 |
| | 2012 | 4 | 0 | 4 | 4 | 0 | 0 |
| | 2013 | 2 | 0 | 2 | 2 | 0 | 0 |
| Dauphin Island | 2010 | 6 | 6 | 0 | 0 | 0 | 6 |
| Barataria Bay | 2010 | 4 | 3 | 2 | 1 | 0 | 3 |
| | 2013 | 5 | 5 | 1 | 4 | 2 | 2 |

**Table 2. Specimens and digestive tracts examined.** n—number of specimens. Metaplasia indicates the number of specimens that exhibited some form of metaplasia, including the absence of cilia on digestive tract diverticula, and whether those diverticula were atrophied and/or possessed heavily vacuolated tissues.

| Locality | Year | n | Metaplasia | Ciliated | Diverticula | |
|---|---|---|---|---|---|---|
| | | | | | atrophied | vacuolated |
| Apalachicola | 2011 | 2 | 2 | 0 | 2 | 2 |
| | 2013 | 7 | 6 | 3 | 4 | 1 |
| Chesapeake | 2010 | 1 | 0 | 1 | 0 | - |
| | 2012 | 4 | 0 | 4 | 0 | 0 |
| | 2013 | 3 | 1 | 3 | 0 | 1 |
| Barataria Bay | 2010 | 5 | 4 | 1 | 4 | 2 |
| | 2013 | 5 | 2 | 2 | 2 | 0 |

## Histology

Shells were opened by severing the adductor muscle as close to the right valve as possible. All specimens were examined after thawing and prior to histological preparation. Any specimen with visible ice crystals within the shell or on the soft body were discarded and excluded from the study. A macroscopic examination was then conducted of the internal surfaces of both valves and the soft tissues. Valves were examined for boring sponges, blisters, and cysts, while the soft tissue was examined for cysts and macroscopically visible parasites. Thirty eight specimens were determined to be suitable for further histological analysis, being free of ice, macroscopic abnormalities and commensal or macro-parasitic organisms. Soft tissue heights ranged between 4-9 cm (average and standard deviation = 6.4±1.26 cm).

Gills were dissected from the body and transverse cuts made through the digestive system. Both the gills and digestive tissues were then immersed in Bouin's fixative for 24 hours, after which they were washed in 50% alcohol for two hours and then stored in 70% ethanol for 24-48 hours. Tissue samples were subsequently dehydrated under vacuum in alcohol solutions of increasing concentration: 80% ethanol for five minutes, two changes of 95% ethanol for 10 minutes and 20 minutes respectively, and then four changes of 100% ethanol for times ranging from 15 to 30 minutes. Following dehydration, specimens were cleared in two baths of xylene, under vacuum, for 15 and 20 minutes respectively, and placed in baths for 30 minutes consisting of equal parts xylene and paraffin. Finally, specimens were infiltrated with pure paraffin from which block molds were made, and subsequently refrigerated for 24 hours at 1.6°C.

Tissue sections were sliced from each block with a rotary microtome, with sections ranging between 7-8 $\mu$m in thickness. These sections were mounted on standard microscope slides with paraffin removed using two washes of xylene lasting five minutes each; tissues were re-hydrated with washes of alcohol of decreasing concentration, ranging from 100-30% for a total of five washes. Re-hydration was completed by immersion in distilled water for two minutes. Tissue sections were then stained with hematoxylin and eosin, followed again by dehydration using three washes of 100% alcohol. Slides were finally cleared with a wash of xylene prior to the application of coverslips. Slides were examined immediately after staining with a Leica DM750 microscope and images taken. More detailed examinations and microphotographs were subsequently made with a Spot Flex Ultra High Resolution CCD camera mounted on a Leica DMRB microscope.

A collecting permit was obtained for Grand Isle from the Louisiana Department of Wildlife and Fisheries (Permit S-121-OYS-2010). A permit was not necessary for collecting on Dauphin Island, and specimens from Apalachicola and Chesapeake Bays were obtained from commercial sources. All figured slides are deposited in the collections of the Department of

Invertebrate Zoology and Geology at the California Academy of Sciences (catalog numbers CASIZ 234232-234238). The corresponding shells are deposited in the collections of the South Dakota School of Mines and Technology (catalog numbers SDSM 157561-157567).

## Results

It was not possible in all cases to examine all ctenidial and digestive tissues in the same specimen, due to the delicacy of the tissues and resulting losses during histological preparation. Both tissues were examined, however, for 18 specimens distributed among all localities (Tables 1 and 2). No indication of an association between the occurrence of metaplasia in ctenidial tissues and their occurrence in digestive tissues was found; we tested this observation with a binary association between the presence/absence of metaplasia in the ctenidia and digestive tract (Fisher's exact test, $p = 0.366$). We therefore considered each tissue separately in the following analyses, testing for effects among all localities, among localities within the GoM, and among years of collection within each locality.

Specimens from the GoM were frozen for periods ranging from one to 16 months (average = 6.2 months), and control specimens from Chesapeake Bay were frozen between one and eight months (average = 2.5 months). We tested for the dependence of the occurrence of any type of metaplasia on the duration for which a specimen was frozen prior to histological analysis, and rejected any such dependence for both ctenidial and digestive tissues (Logistic regression: ctenidia, $\chi^2 = 0.25$, $p = 0.617$; digestive, $\chi^2 = 1.97$, $p = 0.161$). All GoM specimens exhibited some type of ctenidial metaplasia regardless of the duration of freezing, whereas no specimens from Chesapeake Bay did. GoM specimens exhibited the highest frequency (60%) of digestive tract metaplasia from a single cohort, those collected from Grand Isle in August 2013, and frozen for two months prior to analysis. Specimens frozen for seven and 16 months comprise the remaining specimens exhibiting digestive tract metaplasia, but metaplasia was absent in specimens frozen for three, seven and 10 months. A single specimen from Chesapeake Bay exhibited digestive tract metaplasia, and it was frozen for three months prior to analysis. We therefore conclude that the duration of freezing prior to analysis did not introduce tissue artifacts that would otherwise bias the following results.

### Ctenidia

Metaplasia was manifested in ctenidia as the absence of cilia or alteration of the columnar epithelium, or both, with a significant association between the two (Fisher's exact test, $p = 0.0003$). Twenty-one of 23 specimens lacking cilia also exhibited epithelial metaplasia, whereas nine of 13 ciliated specimens had normal, columnar epithelia (Fig 2A). The frequency of ctenidial metaplasia varied significantly among localities, with 100% of specimens from Apalachicola Bay and Dauphin Island exhibiting some degree of alteration, 89% from Louisiana, but none from Chesapeake Bay (n = 37, Fisher's exact test, $p < 0.0001$) (Fig 1; Table 1). There were no significant differences among GoM localities (n = 30, Fisher's exact test, $p = 0.5$).

Metaplasia of the cilia, defined as the absence of cilia, and exclusive of epithelial condition, varied significantly among localities solely because of the higher frequency of metaplasia in GoM samples compared to those from Chesapeake Bay (n = 37, Fisher's exact test, $p = 0.0011$) (Fig 2B); there was no significant difference among GoM localities when tested separately from Chesapeake Bay (n = 30, Fisher's exact test, $p = 0.2942$) (Fig 3). Apalachicola Bay and Louisiana samples were heterogeneous, however, having both ciliated and unciliated individuals, whereas all specimens from Dauphin Island lacked cilia. We tested for a temporal signal among collection years, comparing 2010 and 2013 for Louisiana, and 2010, 2011 and 2013 for

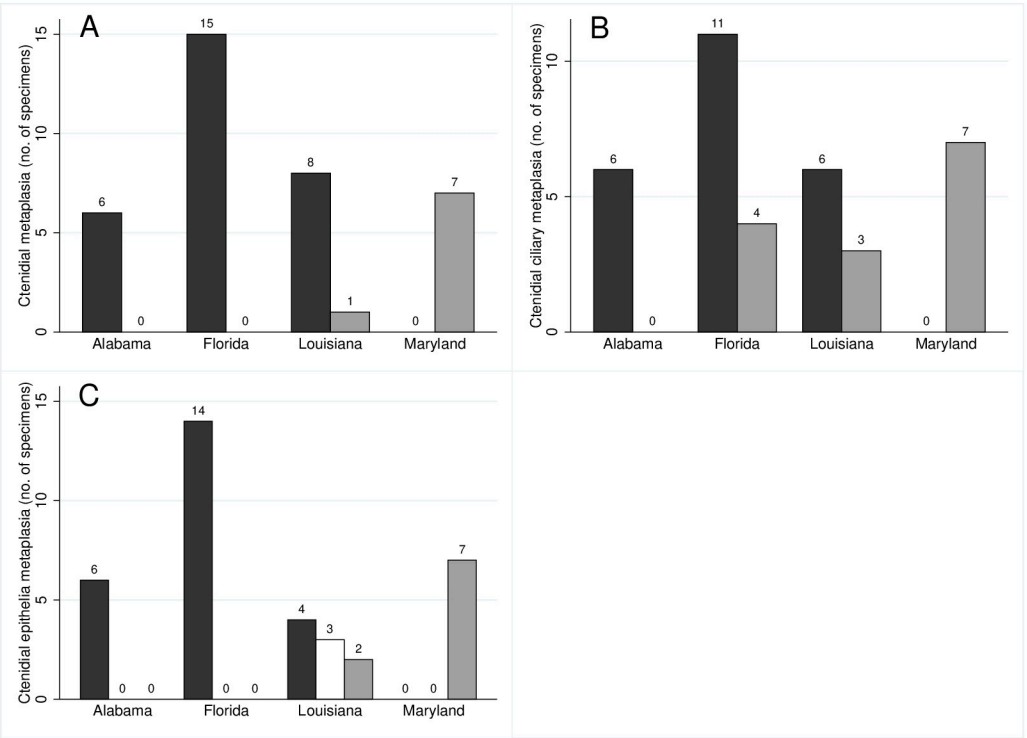

**Fig 2. Numbers of specimens displaying ctenidial metaplasia.** A—Occurrence of any type of ctenidial metaplasia (black) versus non-metaplasial individuals (grey). B—Absence of ctenidial cilia (black) versus presence (grey). C—Alteration of ctendial epithelia (black), healthy epithelia (grey), and mixed altered and healthy in the same individual (white). Locations are Barataria Bay (Louisiana), Dauphin Island (Alabama), Apalachicola (Florida) and Chesapeake Bay (Maryland). Numbers above bars indicate the number of specimens. The results are summarized for all sampled years.

Apalachicola Bay. There was no difference between years for Louisiana (n = 9, Fisher's exact test, $p$ = 0.5328), but significant variation among years in Apalachicola Bay (n = 15, Fisher's exact test, $p$ = 0.0015). One of 12 specimens collected from Apalachicola Bay in 2010, and another in 2011, were ciliated, whereas all three specimens collected and analyzed in 2013 were ciliated.

The presence of metaplasia in gill epithelia differed significantly among all localities (n = 36, Fisher's exact test, $p < 0.0001$), and among GoM localities only (n = 29, Fisher's exact test, $p$ = 0.0011) (Fig 2C). All specimens from Chesapeake Bay had the expected columnar epithelia (Fig 1), but specimens from the GoM exhibited a variety of metaplasial alterations, such as stratified squamous epithelia (Fig 3), and hyperplasia, a condition where the density of cells in ctenidial branches increases dramatically. Several individuals from Louisiana possessed a mixture of normal columnar and stratified squamous epithelia, and one specimen from Apalachicola Bay had mixed cuboidal and stratified squamous epithelia. Differences among the GoM localities is caused by the presence of several individuals from Louisiana with normal, columnar epithelia, whereas all individuals from Apalachicola Bay and Dauphin Island had altered epithelia. There is no evidence to support a temporal difference in Louisiana between years 2010 and 2013, however, and individuals with normal epithelia occur in both years alongside individuals with non-columnar epithelia (n = 9, Fisher's exact test, $p$ = 0.5238).

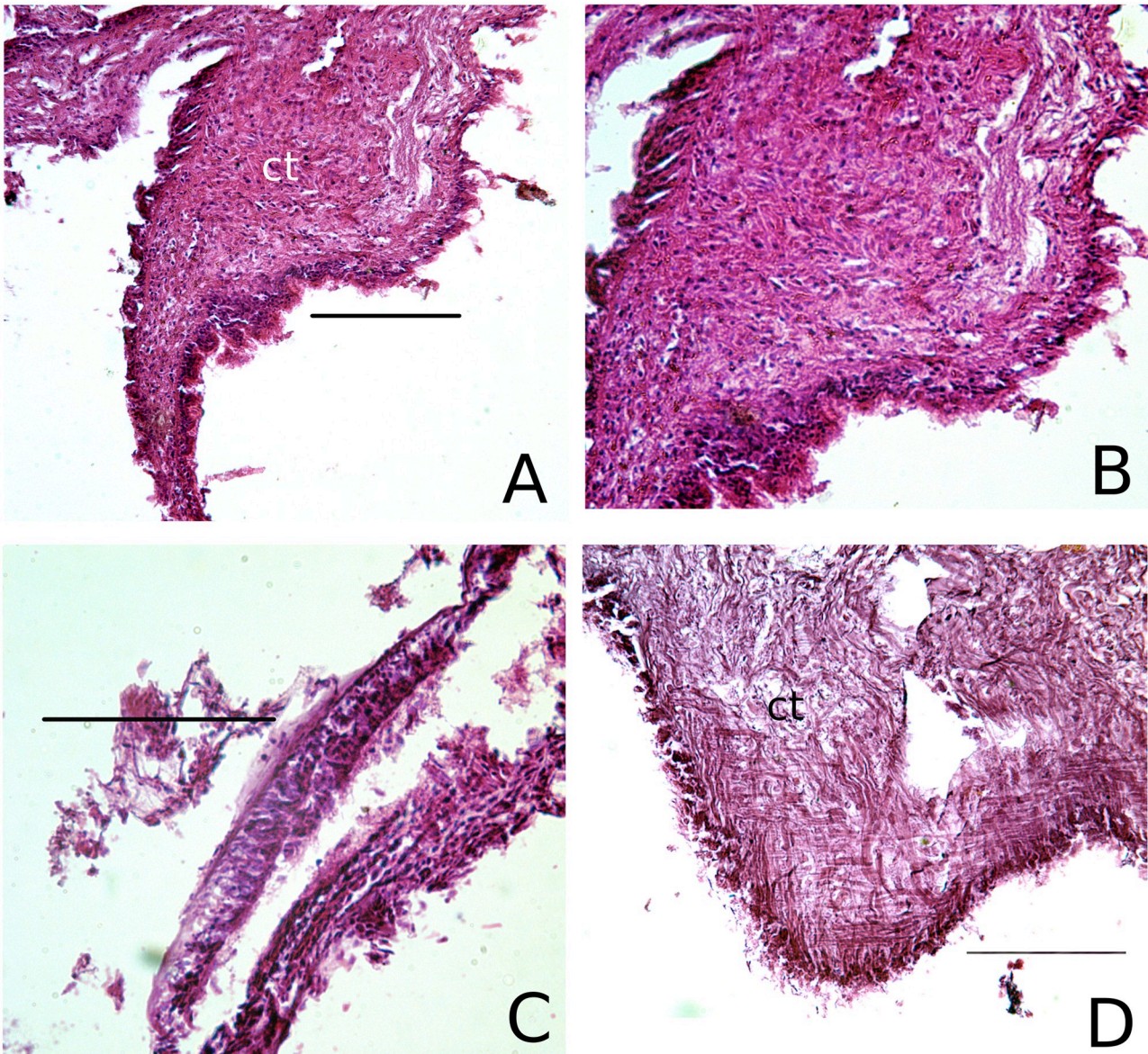

**Fig 3. Metaplasially altered ctenidia of three individuals (A, B—Apalachicola; C—Apalachicola; D—Barataria Bay) from the Gulf of Mexico.** A, B (a magnification of A) and D exhibit hyperplasia of the connective tissue and reduced epithelial structure. C exhibits stratified epithelium and a reduction of cilia on an otherwise developed ctenidial branch. Labels: ct—connective tissue. Scale bars in all images are 100$\mu$m.

### Digestive tract

Three general categories of metaplasia were observed in the digestive tract: the absence of epithelial cilia, atrophy of gut diverticula, and extensively vacuolated cells (Fig 4) (Table 2). The absence of cilia was significantly associated with atrophy of gut diverticula (Fisher's exact test, p = 0.018), and atrophied diverticula were significantly associated with cell vacuolation (Fisher's exact test, p = 0.010). The occurrence of any type of metaplasia differed significantly among localities (Chesapeake Bay, Apalachicola Bay, Louisiana; no digestive system tissues were successfully sectioned for specimens from Dauphin Island) (Fisher's exact test, n = 27,

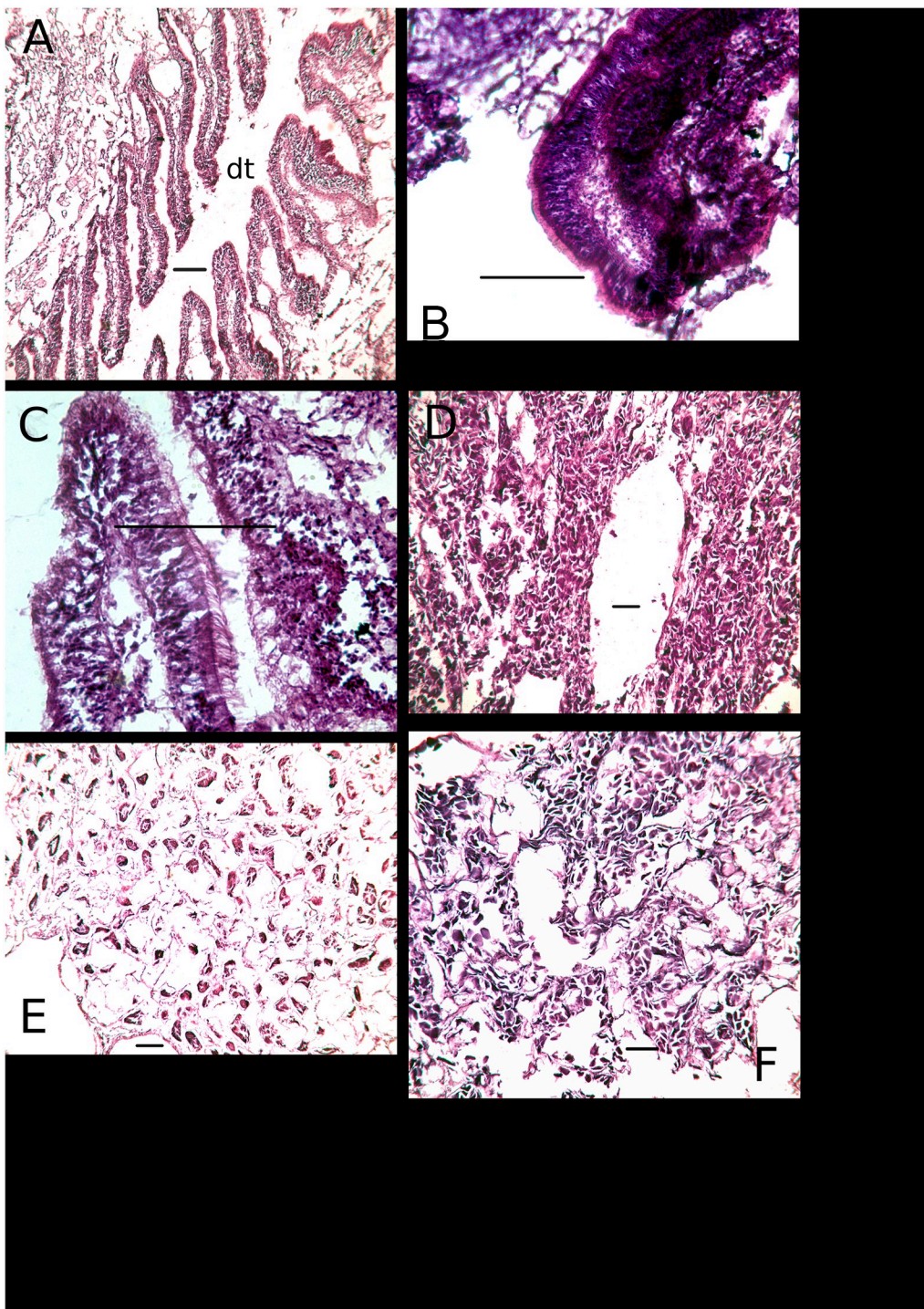

**Fig 4. Digestive tract histologies.** A-C, Normal digestive diverticula of three individuals from Chesapeake Bay. A illustrates the digestive tract (dt) and diverticular branches, and B and C exhibit ciliated columnar epithelia lining the diverticula. exhibiting ciliated columnar epithelia. D-F, three individuals from Barataria Bay, Louisiana, displaying atrophy and vacuolization of tissue resulting in breakdown of epithelial integrity. D—digestive tract with atrophied diverticula. E—epithelium along the digestive tract showing a loss of epithelial integrity. F—A digestive tract consisting of multiple atrophied tubules. Scale bars in all images are 100$\mu$m.

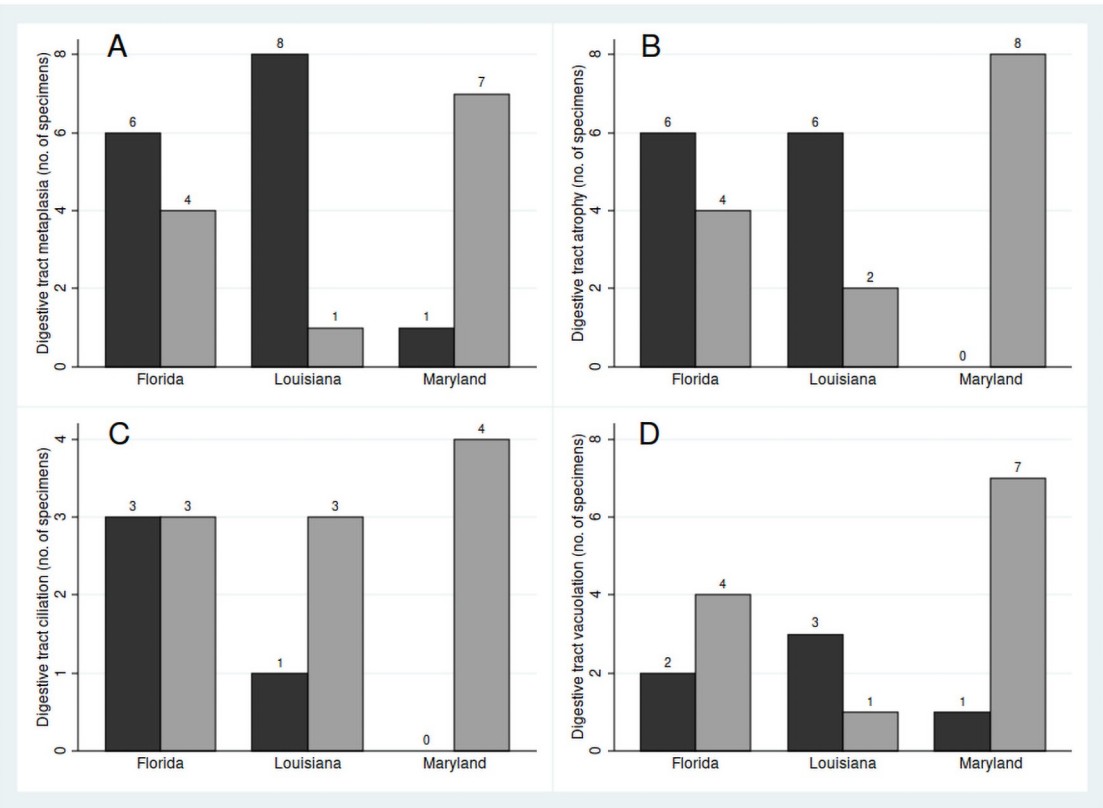

**Fig 5. Proportions of specimens displaying digestive tract metaplasia.** A—Occurrence of any type of digestive metaplasia (black) versus healthy individuals (grey). B—Atrophy of tract diverticula (black) versus normal diverticula (grey). C—Absence of digestive tract cilia (black), versus ciliated tracts (grey). D—Extensive cellular vacuolation (black), versus non-vacuolated cells (grey). Locations are Barataria Bay (Louisiana), Apalachicola (Florida) and Chesapeake Bay (Maryland). Numbers above bars indicate the number of specimens.

p = 0.0068), and is accounted for by a significantly higher frequency in the GoM, where the two localities did not differ from each other (Fisher's exact test, n = 18, p = 0.303) (Fig 5). Furthermore, the difference between Chesapeake Bay and GoM localities results solely from a higher frequency of atrophied diverticula in GoM specimens (Fisher's exact test, n = 26, p = 0.0048), and there are no significant differences with respect to the presence of cilia (Fisher's exact test, n = 14, p = 0.293) or cell vacuolation (Fisher's exact test, n = 17, p = 0.165). Finally, the occurrence of individuals with atrophied diverticula or vacuolated cells, in years 2010, 2011 and 2013 in Apalachicola Bay and Louisiana, renders insignificant any apparent trends in a reduction of metaplasia over that interval (Fisher's exact test, n = 18, p = 0.354; n = 10, p = 0.080 respectively).

## Discussion

We found that specimens of *Crassostrea virginica* from Chesapeake Bay display ctenidial and digestive tissue morphologies and histologies consistent with those described previously for molluscan bivalves in general, and ostreid bivalves specifically. The frequency of ctenidial metaplasia was significantly greater in the Gulf of Mexico than in Chesapeake Bay during the interval covered by this study, but no significant differences among localities within the GoM

were found. Ctenidial metaplasia included the absence of epithelial cilia and alteration of the expected columnar epithelium to cuboidal or squamous cells, or a combination of all those cell types. A lack of cilia was always associated with some type of epithelial alteration, but epithelial cell type alteration could occur without a corresponding lack of cilia. Furthermore, variation in ciliary metaplasia is indicated; all specimens from Alabama were unciliated, collections from Louisiana always contained some ciliated individuals, regardless of the year in which they were collected, whereas collections from Florida showed a significantly declining incidence of metaplasia during the study, that is, an increasing frequency of healthy, ciliated individuals. Whether this indicates a reversal of the conditions that induced ciliary metaplasia in the first place would have to be determined on the basis of continued monitoring.

Metaplasial variation of the ctenidial epithelium was complex both within and between individuals from the GoM, but some form of metaplasia was present in specimens from all localities. Both the cuboidal and squamous alterations represent a decrease in the height of the epithelia compared to normal columnar cells, and in most cases represented a partial to complete collapse of ctenidial branches. A qualitative conclusion drawn from our examination of the altered tissues is that both the absence of columnar cells, replaced instead by cuboidal or squamous cells, and the morphology of the altered branches, result in tissues that bear a strong resemblance to the ectodermal ridges from which the ctenidial structure develops ontogenetically. We therefore speculate that metaplasia in this case could consist of, or include a reversion to an ontogenetically earlier developmental morphology. Furthermore, in contrast to the ciliary condition, there is no indication of temporal variation or a reversal of metaplasia in the epithelial tissues during the sampling interval.

Both the digestive tract and stomach exhibited significant metaplasia in specimens from the Gulf of Mexico, whereas specimens from Chesapeake Bay differed little or not at all from the expected morphologies. There was no clear association of metaplasia between the ctenidia and digestive system in our specimens, however; metaplasia of ctenidial tissues was not predictive of metaplasia of digestive tissues and vice versa. The lack of association points to a complex relationship between the stressors that induce metaplasia and the ways in which individuals may respond.

Digestive tissues exhibited a similar array of metaplasial conditions, comprising an absence of cilia, atrophy of the digestive tract, and vacuolation of gastric tissues. Specimens from the GoM displayed a significantly higher frequency of metaplasia compared to those from Chesapeake Bay, and this was due solely to the occurrence of digestive tract atrophy in the GoM, a condition which was completely absent in Chesapeake Bay specimens. Additionally, the absence of cilia and tissue vacuolation, although both significantly associated with digestive tract atrophy, were also present in some specimens from Chesapeake Bay. Finally, there was no evidence of a declining incidence of metaplasia in GoM specimens during the sampling interval (2010-2013).

Metaplasial tissues may be expected to perform physiologically differently from unaltered tissues, and their presence has implications for individual and population performance [55]. An overall implication, therefore, of metaplasia of respiratory and digestive tissues is a decreased physiological performance of individuals. For example, cilia on ctenidial branches are responsible for particle capture and sorting during filter feeding, moving food particles toward the labial palps for subsequent ingestion. The ctenidia themselves feature significantly in gas exchange, and squamous or cuboidal epithelia form ctenidia with less surface area for efficient gas exchange in comparison to those constructed from columnar epithelia. Similarly, atrophied digestive tracts have reduced surface area for the release of digestive enzymes and the subsequent absorption of nutrients. It is also possible that the composition of the suite of digestive enzymes secreted during the digestive process is altered as a consequence of atrophy. A reduction or absence of cilia within the digestive tract would result in reduced movement of ingested

materials through the tract, and possibly hinder egestion. We speculate that the vacuolation of digestive tissues has a negative impact on digestive and absorptive surface area, as well as the production and secretion of enzymes. It has been established that diseased, stressed or non-feeding bivalves, including *C. virginica*, possess metaplasial digestive tract epithelia [55–57].

The frequency of metaplasia documented in this study would suggest first, that specimens from the GoM perform more poorly in comparison to those from Chesapeake Bay, and second that performance perhaps varied during the sampling interval of the study, with respiratory performance improving over time. If these results can be extrapolated to the populations which were sampled, then one would expect a decline of population productivity in the GoM between 2010 and 2013. This expectation is based both on physiological performance, as well as aspects of population dynamics, such as reproductive output, because of the involvement of physiological systems in reproduction. For example, although species of *Crassostrea* are broadcast spawners, the ctenidial and inhalant-exhalant water systems are involved in the internal movement and spawning of fertilized gametes [58].

An outstanding question that motivated this study is if the Deepwater Horizon (DWH) oil spill, and mitigation measures such as dispersants, can be implicated in the types and frequencies of metaplasia observed [38]. Evidence in support of such a suggestion are the significantly higher frequencies observed in the GoM compared to Chesapeake Bay, and perhaps the observed reduction of ctenidial metaplasia during the interval following the spill. If this is indeed the case, it should be noted that the specimens used for this study were not directly contaminated with DWH oil, even the August 2010 Grand Isle, Louisiana specimens. Therefore the impacts of spills may not have to result from direct contact with a fresh, undiluted slick and can still be measured months to years after initial contamination [21, 59–64].

There are, however, several difficulties with establishing support for this hypothesis. First, no baseline exists for the occurrence and types of metaplasia in the areas sampled. Whereas it is feasible that variation of metaplasial type and frequencies are the result of variations of exposure to DWH oil, it is equally likely given current evidence that metaplasia within the GoM is the result of either exposure to naturally occurring hydrocarbons (from natural seeps), or that it is a now common feature within the GoM because of the long history of petroleum exploration, production and contamination there. An alternative hypothesis is that our inference of metaplasia in the GoM oysters is incorrect, and the tissue anomalies are in fact the result of evolutionary change driven by long-term exposure of the GoM population to both naturally occurring and anthropogenically derived hydrocarbons. A similar explanation was offered for the occurrence of abnormal tissues in bivalves sampled from within oil fields off Trinidad [49]. This explanation could be consistent with the fact that our specimens from Chesapeake Bay do not exhibit the abnormalities so common in the GoM, and would therefore be the result of much greater exposure to hydrocarbons in the GoM, both historically and because of the Deepwater Horizon spill. Alternatively, given the reportedly great sensitivity of oysters in the Chesapeake system to chronic exposure to pollutants, it remains unclear why our specimens from that population are significantly healthier than specimens from the GoM, unless the former is a positive benefit of ongoing mitigation efforts in the Chesapeake system. Resolving these questions requires more spatially and temporally extensive sampling and monitoring both within geographic areas of concern, and areas considered to have fewer or distinct anthropogenic impacts.

## Acknowledgments

Field work in Louisiana and Alabama was assisted by Annette Engel, Carrol Michael, and Caroline Dietz. Timothy Chung, Michael Hellman and Scott Elliott assisted with histological preparations.

## Author Contributions

**Conceptualization:** Deanne S. Roopnarine, Peter D. Roopnarine, Laurie C. Anderson.

**Data curation:** Peter D. Roopnarine, Laurie C. Anderson.

**Formal analysis:** Deanne S. Roopnarine, Peter D. Roopnarine.

**Funding acquisition:** Deanne S. Roopnarine, Laurie C. Anderson.

**Investigation:** Deanne S. Roopnarine, Peter D. Roopnarine, Laurie C. Anderson, Ji Hae Hwang, Swati Patel.

**Methodology:** Deanne S. Roopnarine, Ji Hae Hwang, Swati Patel.

**Project administration:** Peter D. Roopnarine.

**Resources:** Peter D. Roopnarine, Laurie C. Anderson, Swati Patel.

**Supervision:** Deanne S. Roopnarine.

**Visualization:** Peter D. Roopnarine, Ji Hae Hwang, Swati Patel.

**Writing – original draft:** Deanne S. Roopnarine, Peter D. Roopnarine, Laurie C. Anderson, Ji Hae Hwang, Swati Patel.

**Writing – review & editing:** Deanne S. Roopnarine, Peter D. Roopnarine, Laurie C. Anderson.

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
