## [Decision Letter · Decision Letter 0]

26 Mar 2021

PONE-D-21-04744

Metaplasia of respiratory and digestive tissues in the Eastern oyster *Crassostrea virginica* associated with the Deepwater Horizon oil spill

PLOS ONE

Dear Dr. Roopnarine,

Thank you for submitting your manuscript to PLOS ONE. After careful consideration, we feel that it has merit but does not fully meet PLOS ONE’s publication criteria as it currently stands. Reviewers pointed out that the freezing procedure could be a potential source of fatal errors. A proper justification for the procedure is required. Therefore, we invite you to submit a revised version of the manuscript that addresses the points raised during the review process.

We look forward to receiving your revised manuscript.

Kind regards,

Edmond Sanganyado, Ph.D.

Academic Editor

PLOS ONE

Journal Requirements:

2. In your Methods section, please provide additional location information of the collection sites, including geographic coordinates for the data set if available.

Reviewers' comments:

Reviewer's Responses to Questions

**Comments to the Author**

1. Is the manuscript technically sound, and do the data support the conclusions?

Reviewer #1: No

Reviewer #2: Partly

2. Has the statistical analysis been performed appropriately and rigorously? 

Reviewer #1: Yes

Reviewer #2: Yes

3. Have the authors made all data underlying the findings in their manuscript fully available?

Reviewer #1: Yes

Reviewer #2: No

4. Is the manuscript presented in an intelligible fashion and written in standard English?

Reviewer #1: Yes

Reviewer #2: Yes

5. Review Comments to the Author

Reviewer #1: The manuscript titled Metaplasia of respiratory and digestive tissues in the Eastern oyster Crassostrea virginica associated with the Deepwater Horizon oil spill assessed the effects of Deepwater oil spills on tissues of wild adult oysters from a site polluted by an oil spill (northern Gulf of Mexico) and compared them to geographic controls. This work is important because it builds our understanding on the biological and ecological effects of oil spills. The article is generally well written. However, I have a few queries which I list below:

Oyster samples were initially placed on ice during sampling and later stored at -18oC to -20oC until histological analysis with storage times varying between 3 days to 12 months (Line 201). Tissues meant for histological analysis are generally stored in 70% alcohol or in formalin. This is because freeing can damage and alter tissues/cells. Expansion of water due to freezing stretches and damages cell membranes. Some of the effects of freezing are loss of epithelial cilia and intracellular vacuolation of cells, which are also some of the reported effects in this manuscript. See Baraibar MA, Schoning P. Effects of freezing and frozen storage on histological characteristics of canine tissues. J Forensic Sci. 1985 Apr;30(2):439-47. The different storage times 3 days-12 months are particularly problematic because the damage caused by freezing increases with duration of freezing (see https://doi.org/10.1016/S0379-0738(99)00043-2) . If the authors preferred freezing, an alternative would have been fast freezing in liquid nitrogen, and subsequent storage at -80 degrees to avoid formation of tissue-damaging ice crystals. Unless the authors can provide supporting literature to validate their method, this is a fatal error which can seriously affect the findings.

The introduction needs to be streamlined and shortened, particularly after page 3.

Figure 2 are colour bar graphs in red and blue. I suggest these be changed or modified to include patten fill so as to improve accessibility for the visually impaired, particularly the colour blind.

The micrographs Figure 1,3,4 have a low resolution. It will help improve the quality if the authors have images of higher quality.

Line 34: correct disperants to dispersants

Reviewer #2: General comments

This manuscript describes metaplasia of oyster tissues and their association with the Deepwater Horizon oil spill. Although it is interesting, the main concern would be the very low number of specimens analyzed (ranged from 1 to 9).

The authors also failed to mention if these specimens were of the same size or life stages – different life stages might exhibit different response or degree of metaplasia towards oil spill; and if water parameters were taken during sampling – this could provide a stronger correlation between the present and level of oil in water column with the occurrence of metaplasia.

Since the authors mentioned that metaplasia could be present years after a spill (line 127-138), I would expect the authors to include samples after half a decade and current year as well?

Also, the introduction section is too long. Kindly revise.

Line 140 & 143: ‘et al.’

Line 239: The authors mentioned in the Method section, only 38 samples were used, not 46?

Line 284: Figure ??

For all histological figures, please standardize the placement of scale bar to only lower right of each figure.

6. PLOS authors have the option to publish the peer review history of their article (what does this mean?). If published, this will include your full peer review and any attached files.

Reviewer #1: **Yes: **Charles Teta

Reviewer #2: **Yes: **Khor Waiho

---

## [Author Response · Author response to Decision Letter 0]

7 Jul 2021

Responses to reviewers

PONE-D-21-04744

Metaplasia of respiratory and digestive tissues in the Eastern oyster Crassostrea virginica associated with the Deepwater Horizon oil spill

Reviewer #1

“Oyster samples were initially placed on ice during sampling and later stored at -18oC to -20oC until histological analysis with storage times varying between 3 days to 12 months (Line 201). Tissues meant for histological analysis are generally stored in 70% alcohol or in formalin. This is because freeing can damage and alter tissues/cells. Expansion of water due to freezing stretches and damages cell membranes. Some of the effects of freezing are loss of epithelial cilia and intracellular vacuolation of cells, which are also some of the reported effects in this manuscript. See Baraibar MA, Schoning P. Effects of freezing and frozen storage on histological characteristics of canine tissues. J Forensic Sci. 1985 Apr;30(2):439-47. The different storage times 3 days-12 months are particularly problematic because the damage caused by freezing increases with duration of freezing (see https://avanan.url-protection.) . If the authors preferred freezing, an alternative would have been fast freezing in liquid nitrogen, and subsequent storage at -80 degrees to avoid formation of tissue-damaging ice crystals. Unless the authors can provide supporting literature to validate their method, this is a fatal error which can seriously affect the findings.”

We agree with the reviewer and were of this potential problem. As explained in the revised manuscript, the specimens were not originally intended for histological analysis, but instead for chemical analyses. Therefore, soft tissues were not placed in preservatives suitable for histological analysis. Our decision to pursue histological analysis raised the potential issues of freezing, but we addressed this by treating our control specimens from Chesapeake Bay, Maryland, in the same manner as specimens from the Gulf of Mexico, including deep freezing the specimens. Given that those specimens did not exhibit the metaplasia observed in the Gulf of Mexico specimens, we concluded that metaplasia in the latter specimens was not an artifact of freezing. We have explained this in more detail within the manuscript as follows:

Lines 190-198: Freezing as a preservation technique for histological analysis can be problematic

because of the formation of ice crystals at the cellular level, and the consequential

damage of tissues. Specimens for this study were frozen because they were originally

intended for chemical analysis of soft and hard tissues, specifically the measurement of

heavy metal concentrations [40]. Thus chemical preservation was avoided and freezing

employed instead. We tested for the possible introduction of histological artifacts due to

freezing, and the possibility of attributing those incorrectly to metaplasia, by freezing

the control specimens from Chesapeake Bay at temperatures and for durations

comparable to those used for the GoM specimens.

Lines 248-263: Specimens from the GoM were frozen for periods ranging from one to 16 months

(average = 6.2 months), and control specimens from Chesapeake Bay were frozen

between one and eight months (average = 2.5 months). We tested for the dependence of

the occurrence of any type of metaplasia on the duration for which a specimen was

frozen prior to histological analysis, and rejected any such dependence for both ctenidial

and digestive tissues (Logistic regression: ctenidia, χ2 = 0.25, p = 0.617; digestive,

χ2 = 1.97, p = 0.161). All GoM specimens exhibited some type of ctenidial metaplasia

regardless of the duration of freezing, whereas no specimens from Chesapeake Bay did.

GoM specimens exhibited the highest frequency (60%) of digestive tract metaplasia

from a single cohort, those collected from Grand Isle in August 2013, and frozen for two

months prior to analysis. Specimens frozen for seven and 15 months comprise the

remaining specimens exhibiting digestive tract metaplasia, but metaplasia was absent in

specimens frozen for three, seven and 10 months. A single specimen from Chesapeake

Bay exhibited digestive tract metaplasia, and it was frozen for three months prior to

analysis. We therefore conclude that the duration of freezing prior to analysis did not

introduce tissue artifacts that would otherwise bias the following results.

“The introduction needs to be streamlined and shortened, particularly after page 3.”

We eliminated sections of the Introduction and shortened it overall, as can be followed in the tracked changes version of the manuscript.

“Figure 2 are colour bar graphs in red and blue. I suggest these be changed or modified to include patten fill so as to improve accessibility for the visually impaired, particularly the colour blind.”

We have modified both Figure 2 and the other bar graph figure so that they are no longer coloured, but instead use black, white and grey solid bars. The contrast is therefore increased and colour removed.

“The micrographs Figure 1,3,4 have a low resolution. It will help improve the quality if the authors have images of higher quality.”

The micrographs are all of a minimum resolution of 300 dpi (or higher) as required by PLoS, and thus meet the journal’s specifications. Several of the higher magnification images were made under oil immersion, perhaps giving the impression that they are of low resolution, but they are not.

“Line 34: correct disperants to dispersants”

Corrected.

Reviewer #2

“The authors also failed to mention if these specimens were of the same size or life stages – different life stages might exhibit different response or degree of metaplasia towards oil spill; and if water parameters were taken during sampling – this could provide a stronger correlation between the present and level of oil in water column with the occurrence of metaplasia.”

Specimens were of variable size, but all were of sufficient size where it is believed that the species has attained sexual maturity, as outlined in government collecting guidelines. Water samples were unfortunately not taken when specimens were collected, but at no times were oil slicks visible, as mentioned in the Introduction. We have revised the manuscript to describe both shell and soft tissue sizes of the specimens, as follows:

Lines 186-187: Shells ranged in height between 8-12 cm, ensuring that all individuals were post-juveniles.

Lines 208-209: Soft tissue heights ranged between 4-9 cm (average and standard deviation = 6.4±1.26 cm).

“Since the authors mentioned that metaplasia could be present years after a spill (line 127-138), I would expect the authors to include samples after half a decade and current year as well?”

Unfortunately this was not possible. Collecting after 2013 became logistically difficult for the research team because no authors were in close proximity to any of the sampling sites. Nevertheless, we planned to take advantage of the extensive collecting undertaken by the United States federal government during that time period. When the specimens were made available to researchers in 2017 we applied for, and received more than 100 specimens from various locations in the Gulf of Mexico. Unfortunately, none of those specimens had been preserved properly, being mostly dessicated and there unsuitable for histological analysis.

“Line 140 & 143: ‘et al.’”

Corrected.

“Line 239: The authors mentioned in the Method section, only 38 samples were used, not 46?”

The correct number is 38, and we have made that consistent throughout the manuscript.

“For all histological figures, please standardize the placement of scale bar to only lower right of each figure.”

This is not possible. Scale bars are placed automatically by the calibration and imaging software of the microscope’s camera for most of the images, and moving the bar would entail altering/editing the image, which we declined to do.

---

## [Decision Letter · Decision Letter 1]

17 Aug 2021

Metaplasia of respiratory and digestive tissues in the Eastern oyster *Crassostrea virginica* associated with the Deepwater Horizon oil spill

PONE-D-21-04744R1

Dear Dr. Roopnarine,

We’re pleased to inform you that your manuscript has been judged scientifically suitable for publication and will be formally accepted for publication once it meets all outstanding technical requirements.

Kind regards,

Edmond Sanganyado, Ph.D.

Academic Editor

PLOS ONE

Additional Editor Comments (optional):

Reviewers' comments:

Reviewer's Responses to Questions

**Comments to the Author**

1. If the authors have adequately addressed your comments raised in a previous round of review and you feel that this manuscript is now acceptable for publication, you may indicate that here to bypass the “Comments to the Author” section, enter your conflict of interest statement in the “Confidential to Editor” section, and submit your "Accept" recommendation.

Reviewer #1: (No Response)

Reviewer #2: (No Response)

2. Is the manuscript technically sound, and do the data support the conclusions?

Reviewer #1: Yes

Reviewer #2: Partly

3. Has the statistical analysis been performed appropriately and rigorously? 

Reviewer #1: Yes

Reviewer #2: Yes

4. Have the authors made all data underlying the findings in their manuscript fully available?

Reviewer #1: No

Reviewer #2: Yes

5. Is the manuscript presented in an intelligible fashion and written in standard English?

Reviewer #1: Yes

Reviewer #2: Yes

6. Review Comments to the Author

Reviewer #1: The authors have now addressed most of my concerns regarding the use of frozen samples, which could introduce artefacts caused by tissue damage resulting from ice crystals. I applaud the authors for being forthright about the potential effects of freezing and carrying out comparisons to frozen control samples as an assurance.

What was the longest duration samples from GOM were frozen? In the Materials and Methods section (page 8, Line 189), the authors state that “storage times varied from 3 days to 12 months”. But under the Results section (page 10, Line 252), it is said that “Specimens from the GoM were frozen for periods ranging from one to 16 months.” I have also seen in one of the sentences suggesting 15 months. The authors need to address this.

Reviewer #2: The authors have addressed most of my concerns. I have just one last suggestion for the improvement of this manuscript. The problem of conducting histological procedure on frozen samples is indeed a major loophole of this study. Although the authors included and compared frozen control specimens from Chesapeake Bay, I would suggest the inclusion of histological comparison of samples treated with normal fixation instead of freezing as well. This will give a more objective comparison, allowing us to know the impact of freezing on oyster tissue, and the minimal impact of freezing on the development of metaplasia.

7. PLOS authors have the option to publish the peer review history of their article (what does this mean?). If published, this will include your full peer review and any attached files.

Reviewer #1: No

Reviewer #2: No

---

## [Editor Report · Acceptance letter]

23 Aug 2021

PONE-D-21-04744R1 

Metaplasia of respiratory and digestive tissues in the Eastern oyster *Crassostrea virginica* associated with the Deepwater Horizon oil spill 

Dear Dr. Roopnarine:

I'm pleased to inform you that your manuscript has been deemed suitable for publication in PLOS ONE. Congratulations! Your manuscript is now with our production department. 

Kind regards, 

on behalf of

Prof. Edmond Sanganyado 

Academic Editor

PLOS ONE